# The interplay of *atoh1* genes in the lower rhombic lip during hindbrain morphogenesis

**Ivan Belzunce, Carla Belmonte-Mateos, Cristina Pujades** *

Department of Experimental and Health Sciences, Universitat Pompeu Fabra, Barcelona, Spain

* cristina.pujades@upf.edu

**Data Availability Statement:** All relevant data are within the paper and its Supporting Information files.

**Funding:** This work was supported by BFU2016-81887-REDT/AEI (MINECO-FEDER), and AEI-

## Abstract

The Lower Rhombic Lip (LRL) is a transient neuroepithelial structure of the dorsal hindbrain, which expands from r2 to r7, and gives rise to deep nuclei of the brainstem, such as the vestibular and auditory nuclei and most posteriorly the precerebellar nuclei. Although there is information about the contribution of specific proneural-progenitor populations to specific deep nuclei, and the distinct rhombomeric contribution, little is known about how progenitor cells from the LRL behave during neurogenesis and how their transition into differentiation is regulated. In this work, we investigated the *atoh1* gene regulatory network operating in the specification of LRL cells, and the kinetics of cell proliferation and behavior of *atoh1a*-derivatives by using complementary strategies in the zebrafish embryo. We unveiled that *atoh1a* is necessary and sufficient for specification of LRL cells by activating *atoh1b*, which worked as a differentiation gene to transition progenitor cells towards neuron differentiation in a Notch-dependent manner. This cell state transition involved the release of *atoh1a*-derivatives from the LRL: *atoh1a* progenitors contributed first to *atoh1b* cells, which are committed non-proliferative precursors, and to the *lhx2b*-neuronal lineage as demonstrated by cell fate studies and functional analyses. Using *in vivo* cell lineage approaches we revealed that the proliferative cell capacity, as well as the mode of division, relied on the position of the *atoh1a* progenitors within the dorsoventral axis. We showed that *atoh1a* may behave as the cell fate selector gene, whereas *atoh1b* functions as a neuronal differentiation gene, contributing to the *lhx2b* neuronal population. *atoh1a*-progenitor cell dynamics (cell proliferation, cell differentiation, and neuronal migration) relies on their position, demonstrating the challenges that progenitor cells face in computing positional information from a dynamic two-dimensional grid in order to generate the stereotyped neuronal structures in the embryonic hindbrain.

## Introduction

The assembly of functional neural circuits requires the specification of neuronal identities and the execution of developmental programs that establish precise neural network wiring. The generation of such cell diversity happens during embryogenesis, at the same time that the brain undergoes a dramatic transformation from a simple tubular structure, the neural tube, to

PGC2018-095663-B-I00 FEDER/UE (MICIU) to CP, and Unidad de Excelencia María de Maetzu MDM-2014-0370 and AEI-CEX2018-000792-M to DCEXS-UPF. IB and CBM were recipients of predoctoral fellowships from the Generalitat de Catalunya (FI) and the Spanish Ministry of Economy, Industry and Competitiveness (FPI), respectively. CP is a recipient of ICREA Academia award (Generalitat de Catalunya). The funders had no role in study design, data collection and analysis, decision to publish, or preparation of the manuscript.

**Competing interests:** The authors have declared that no competing interests exist.

a highly convoluted structure–the brain-, resulting in changes in the position of neuronal progenitors and their derivatives upon time. Thus, the coordination of progenitor proliferation and cell fate specification is central to tissue growth and maintenance.

The comprehension of how neuronal heterogeneity is achieved implies the understanding of how the neurogenic capacity is acquired, how the number of progenitors vs. differentiated neurons is balanced, and how their relative spatial distribution changes upon morphogenesis. Neurogenesis is initiated by proneural genes, which trigger the specification of neuronal lineages and commit progenitors to neuronal differentiation by promoting cell cycle exit and activating a downstream cascade of differentiation genes [1]. Once neuronal progenitors are committed, the first step towards achieving the diversity observed in adults occurs early in development with the division of neuronal progenitor cells into distinct domains along dorsoventral (DV) axis, which will give rise to different types of neurons in response to morphogen signals emanating from local organizing centers [2]. The next level of complexity arises with the interpretation of the two-dimensional grid, along the DV and anteroposterior (AP) axes, of molecularly distinct progenitor regions that will control the final neuronal fate.

The hindbrain undergoes a segmentation process along the AP axis leading to the formation of seven metameres named rhombomeres (r1-r7) that constitute developmental units of gene expression and cell lineage compartments [3–5]. This compartmentalization involves the formation of a cellular interface between segments called the hindbrain boundary [6], which exhibit distinct features such as specific gene expression [7] and biological functions [8–11]. The hindbrain is the most conserved brain vesicle along evolution [12,13], and in all vertebrates the dorsal part of the hindbrain gives rise to a transient neuroepithelial structure, the rhombic lip (RL). RL progenitors will generate different neuronal lineages according to their position along the AP axis. The most anterior region of the RL, which coincides with the dorsal pole of r1, is known as Upper Rhombic Lip (URL) and produces all granule cells of the external and internal granular layers of the cerebellum [14,15]. The rest of the RL, which expands from r2 to r7, is known as Lower Rhombic Lip (LRL) and gives rise to deep nuclei of the brainstem, such as the vestibular and auditory nuclei and most posteriorly the precerebellar nuclei [16,17]. The genetic program for cerebellum development is largely conserved among vertebrates [16]; as an example, zebrafish and mouse use similar mechanisms to control cerebellar neurogenesis with a crucial role of *atoh1* and *ptf1* genes [17,18]. For the LRL, we know both the contribution of *ptf1a/atoh1a* proneural progenitor populations to specific deep nuclei [19], and the distinct rhombomeric identity [20]. However, little is known about how progenitor cells from the LRL behave during neurogenesis and how their transition into differentiation is regulated, in order to balance the rate of differentiation and proliferation to produce the proper neuronal numbers.

In this work, we sought to understand the role of *atoh1* genes in the generation of the neuronal derivatives of LRL. We used complementary strategies in the zebrafish embryos to provide information about the gene regulatory network operating in the specification of LRL cells, and the kinetics of cell proliferation and behavior of *atoh1a*-derivatives. We unveiled that *atoh1a* is necessary and sufficient for specification of LRL cells by activating *atoh1b*, which worked as a differentiation gene to transition progenitor cells towards neuronal differentiation in a Notch-dependent manner. This cell state transition involved the release of *atoh1a*-derivatives from the LRL: *atoh1a* progenitors contributed first to *atoh1b* cells, which are committed non-proliferative precursors, and to the *lhx2b*-neuronal lineage as demonstrated by cell fate studies and functional analyses. Using *in vivo* cell lineage approaches we showed that the proliferative cell as well as their mode of division, relied on the position of the *atoh1a* progenitors within the dorsoventral axis.

## Materials and methods

### Zebrafish lines and genotyping

Zebrafish (*Dario rerio*) were treated according to the Spanish/European regulations for the handling of animals in research. All protocols were approved by the Institutional Animal Care and Use Ethic Committee (Comitè Etica en Experimentació Animal, PRBB) and the Generalitat of Catalonia (Departament de Territori i Sostenibilitat), and they were implemented according to European regulations. Experiments were carried out in accordance with the principles of the 3Rs.

Embryos were obtained by mating of adult fish using standard methods. All zebrafish strains were maintained individually as inbred lines. The transgenic line Mu4127 carries the KalTA4-UAS-mCherry cassette into the 1.5Kb region downstream of *egr2a/krx20* gene, and was used for targeting UAS-constructs to rhombomeres 3 and 5, or as landmark of these regions [21]. Tg[ßactin:HRAS-EGFP] line, called Tg[CAAX:GFP] in the manuscript, displays GFP in the plasma membrane and was used to label the cell contours [22]. Tg[tp1:d2GFP] line is a readout of cells displaying Notch-activity [23] in which cells with active Notch express GFP. The Tg[HuC:GFP] line labels differentiated neurons [24]. Tg[atoh1a:Kalta4;UAS:H2A-mCherry] and Tg[atoh1a:Kalta4;UAS:GFP] fish lines label *atoh1a*-positive cells and their derivatives due to the stability of the fluorescent proteins. They were generated by crossing Tg[atoh1a:Gal4] [25] with Tg[UAS:H2A-mCherry] or Tg[UAS:GFP] lines, respectively, and accordingly were called Tg[atoh1a:H2A-mCherry] and Tg[atoh1a:GFP] all along the manuscript for simplification.

*atoh1a^fh282^* mutant line in the Tg[atoh1a:GFP] background, which carried a missense mutation within the DNA-binding domain, was previously described in [18]. Embryos were phenotyped blind and later genotyped by PCR using the following primers: Fw primer 5′−ATGGA TGGAATGAGCACGGA−3′ and Rv primer 5′−GTCGTTGTCAAAGGCTGGGA−3′. Amplified PCR products underwent digestion with AvaI (New England Biolabs), which generated two bands: 195 bp + 180 bp for the WT allele and 195 bp + 258 bp for the mutant allele. Since the *atoh1a^fh282^* mutant allele only caused a deleterious phenotype in homozygosity, wild type and heterozygous conditions showed identical phenotypes and they were displayed in all our experiments as a single wild type condition.

### Whole mount *in situ* hybridization and immunostainings

Zebrafish whole-mount *in situ* hybridization was adapted from [26]. The following riboprobes were generated by *in vitro* transcription from cloned cDNAs: *atoh1a* and *atoh1b* [27], *ptf1a*, *ascl1a*, *ascl1b* [28], *neurog1* [29], and *neurod4* [30]. *lhx1a* and *lhx2b* probes were generated by PCR amplification adding the T7 promoter sequence in the Rv primers (*lhx2b* Fw primer, 5′− CAG AGA CGA ACA TGC CTT CA−3′; *lhx2b* Rv primer, 5′− ATA TTA ATA CGA CTC ACT ATA CGT CAG GAT TGT GGT TAG ATG −3′; *lhx1a* Fw primer, 5′−CCA GCT ACA GGA CGA TGT CA−3′; *lhx1a* Rv primer, 5′−ATA TTA ATA CGA CTC ACT ATA GAG GGA CGT AAA AGG ACG GAC T−3′). The chromogenic *in situ* hybridizations were developed with NBT/BCIP (blue) substrate. For fluorescent *in situ* hybridization, FLUO- and DIG-labeled probes were detected with TSA Fluorescein and Cy3, respectively.

For immunostaining, embryos were blocked in 5% goat serum in PBS-Tween20 (PBST) during 1h at room temperature and then incubated O/N at 4°C with the primary antibody. The primary antibodies were the following: anti-GFP (1:200; Torrey Pines), anti-pH3 (1:200; Upstate), anti-HuC (1:100, Abcam). After extensive washings with PBST, embryos were incubated with secondary antibodies conjugated with Alexa Fluor®594 or Alexa Fluor®633 (1:500,

**Table 1. Quantification of differentiated cells in *atoh1a^WT^* and *atoh1a^fh282^* embryos at 24hpf and 36hpf with the t-test values (Fig 4M and 4N).**

|  | *atoh1a^WT^* | n | *atoh1a^fh282^* | n | p |
|---|---|---|---|---|---|
| r4/r5-24hpf | 20.5 ± 4 | 14 | 1.4 ± 1.9 | 11 | < 0.001 |
| r5/r6-24hpf | 11.9 ± 3.3 | 14 | 0.25 ± 0.7 | 11 | < 0.001 |
| r4/r5-36hpf | 85.8 ±18.2 | 18 | 26.7 ± 9.5 | 7 | < 0.001 |
| r5/r6-36hpf | 75.6 ± 21.1 | 18 | 25.1 ± 11.9 | 7 | < 0.001 |

Invitrogen). Either Draq5^TM^ (1:2000; Biostatus, DR50200) or DAPI were used to label nuclei. After staining, embryos were either flat-mounted and imaged under a Leica DM6000B fluorescence microscope, or whole-mounted in agarose and imaged under a SP8 Leica confocal microscope.

## BrdU staining and TUNEL analysis

Cells in S-phase were detected by BrdU-incorporation (Roche). Briefly, embryos were dechorionated and incubated in 10mM BrdU diluted in 5%DMSO 30min at RT. Embryos were washed with fresh water, fixed in 4%PFA at RT, and dehydrated in MetOH. After progressive rehydration, embryos were permeabilized with Proteinase K (Invitrogen) at 10 μg/ml 15min at RT, fixed 20min in 4%PFA, and washed 3x10min in PBS before immunostaining with anti-BrdU (1:50, Becton Dickinson).

Distribution of apoptotic cells was determined by TdT-mediated dUTP nick-end labeling of the fragmented DNA (TUNEL, Roche). Briefly, whole embryos at 30hpf were fixed in 4% PFA and dehydrated in 100% MetOH were permeabilized with Proteinase K at 25 μg/ml, and preincubated with TUNEL mixture during 60 min at 37˚C according to the manufacturer's instructions. DAPI (1:500; Molecular Probes) was used to label nuclei.

## Quantification of the phenotypes

For quantifying the number of differentiated neurons in *atoh1a^WT^*Tg[atoh1a:GFP] and *atoh1a^fh282^*Tg[atoh1a:GFP] embryos, confocal MIP of ventral stacks were used and all cells present in the r4/r5 and r5/r6 domain were counted (see Table 1 for numbers and statistics).

In order to quantify the number of proliferating LRL-cells in *atoh1a^WT^* and *atoh1a^fh282^* embryos in the Tg[atoh1a:GFP] background, the number of mitotic figures within the atoh1a: GFP progenitor domain was assessed (see Table 2 for numbers and statistics).

For the quantification of the total number of LRL atoh1a:cells in *atoh1a^WT^* and *atoh1a^fh282^* embryos in the Tg[atoh1a:GFP] background, embryos at 24hpf were stained with Draq5 and the total number of nuclei of atoh1a:GFP cells was assessed in r5 (see Table 2 for numbers and statistics).

For the quantification of the delamination time of atoh1a:cells in *atoh1a^WT^* and *atoh1a^fh282^* embryos in the Tg[atoh1a:GFP] background, we kept track of the time of division of a given cell (t0) and the time of delamination of the resulting cells (tf) and calculated the difference between tf and t0.

**Table 2. Quantification of LRL cells and hallmarks of apoptosis in *atoh1a^WT^* and *atoh1a^fh282^* embryos with the t-test values (Fig 5A–5D).**

|  | *atoh1a^WT^* | n | *atoh1a^fh282^* | n | p |
|---|---|---|---|---|---|
| mitotic atoh1a:GFP LRL cells | 17.9 ± 3.6 | 15 | 15.9 ± 3.1 | 8 | ns |
| total LRL atoh1a:GFP cells | 69.5 ± 6.4 | 15 | 68.4 ± 7.5 | 8 | ns |
| hindbrain apoptotic cells | 10.5 ± 8.1 | 17 | 8.2 ± 4.7 | 5 | ns |

### 3D+time imaging

Double transgenic Tg[atoh1a:H2A-mCherry]Tg[CAAX:GFP] embryos, or $atoh1a^{WT}$Tg [atoh1a:GFP] and $atoh1a^{fh282}$Tg[atoh1a:GFP] embryos were anesthetized and mounted dorsally in 1%LMP-agarose. Time-lapse imaging was performed from 24hpf to 34hpf in a Leica SP8 system using PMT detectors and a 20x objective. Experimental parameters for the videos were: voxel dimension (nm), x416.6 y416.6 z1200; time frame 8 min; total time 14 h; pinhole 1 Airy; zoom 1.3; objective 20x immersion; NA 0.70. The videos were processed and analyzed using Fiji software (NIH). Cell tracking was performed using the MaMuT software (Fiji plug-in) [31].

### Conditional overexpression

The full-length coding sequences of zebrafish *atoh1a-* and *atoh1b* [27] were cloned into the MCS of a custom dual vector that expressed Citrine from one side of 5xUAS sequence and the cDNA of interest from the opposite side [32]. Mu4127 embryos (expressing KalT4 in r3 and r5) were injected either with H2B-citrine:UAS, H2B-citrine:UAS:atoh1a or H2B-citrine:UAS:atoh1b constructs at the one-cell stage, grown at 28.5˚C and analyzed at 24hpf for *atoh1a/b* and *lhxb2 in situ* hybridization and Citrine expression.

### Pharmacological treatments

$atoh1a^{WT}$Tg[atoh1a:GFP] and $atoh1a^{fh282}$Tg[atoh1a:GFP] sibling embryos were treated either with 10 μM of the gamma-secretase inhibitor LY411575 (Stemgent) or DMSO for control. The treatment was applied into the swimming water at 28.5˚C from 24hpf to 30hpf. After treatment, embryos were fixed in 4%PFA for further analysis.

## Results

### Expression of proneural genes within the zebrafish hindbrain

We first analyzed the formation of molecularly distinct neural progenitor domains, each of them able to generate particular neuronal cell types, during hindbrain embryonic development. We performed a comprehensive spatiotemporal analysis of the expression of distinct proneural genes along the anteroposterior (AP) and dorsoventral (DV) axes within the hindbrain and defined the DV order of proneural gene expression. The expression profiles of *atoh1a*, *ptf1a*, *ascl1a*, *ascl1b*, and *neurog1* indicated that their onset of expression differed along the AP axis (S1 Fig). The dorsal most progenitor cells expressed *atoh1a* all along the AP axis from 18hpf onwards, which remained expressed there until at least 48hpf (S1A–S1C Fig; Fig 1A–1E). *ptf1a* expression started in rhombomere 3 (r3) at 18hpf and from 21hpf onwards it expanded anteriorly towards r1 and r2 (S1D and S1E Fig), ending up expressed all along the AP axis of the hindbrain with different intensities (S1F Fig; [17]). These two proneural genes were the most dorsally expressed as shown by transverse sections (S1A'–S1F' Fig). *ascl1a* and *ascl1b* displayed overlapping expression profiles along the AP axis in a rhombomeric restricted manner with slightly different intensities (S1G and S1J Fig). Nevertheless, their DV expression differed: *ascl1a* expression was adjacently dorsal to *ascl1b* and constituted a smaller territory (S1G'–S1I', S1J'–S1L' and S1R Fig). Indeed, *ascl1a* and *ptf1a* mainly overlapped along the DV axis occupying the region in between *atoh1a* and *ascl1b* (S1P–S1R Fig). Although by 24hpf *ascl1a*-cells seemed to be more laterally located than *ascl1b*-cells (compare S1I with S1L Fig), this just reflected the lateral displacement of the dorsal part of the neural tube upon hindbrain ventricle opening: the hindbrain at early stages was a closed neural tube resembling the spinal cord (S1 Fig, 18-21hpf stages), whereas at late stages all progenitor cells were in the ventricular

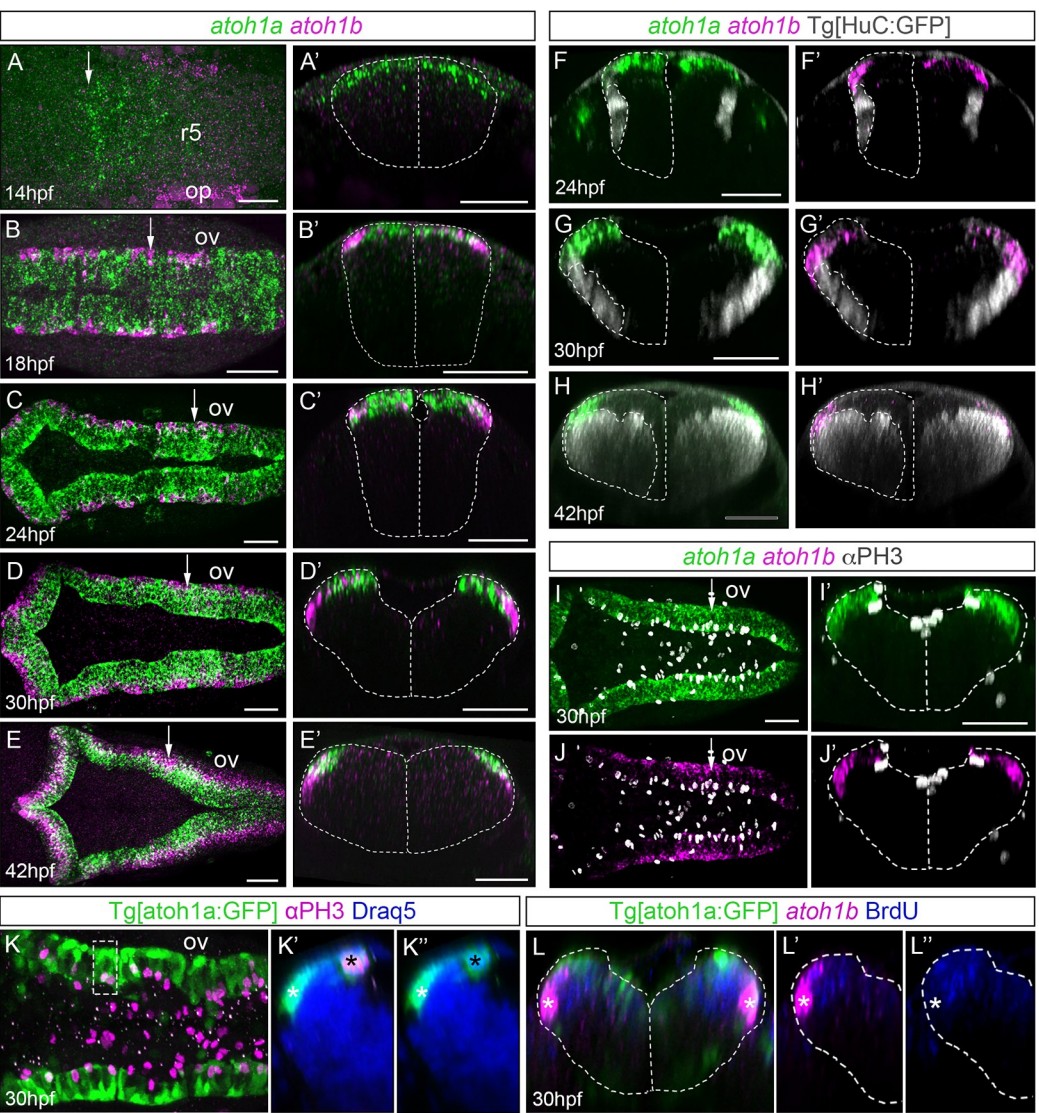

**Fig 1. Spatiotemporal analysis of *atoh1a* and *atoh1b* within the hindbrain.** A-E) Whole mount double *in situ* hybridization with *atoh1a* (green) and *atoh1b* (magenta) in wild type embryos from 14hpf to 42hpf. Dorsal views with anterior to the left. A'-E') Reconstructed transverse views of dorsal views in (A-E) at the level indicated by the white arrow depicted in (A-E). Note that the expression of *atoh1b* is more lateral than *atoh1a*-cells. Dotted line corresponded to the neural tube contour. F-H) Whole mount double *in situ* hybridization with *atoh1a* (green) and *atoh1b* (magenta) on Tg [HuC:GFP] embryos from 24hpf to 42hpf, where HuC expression was displayed in white. Dotted line corresponded to the neural tube and the HuC-expression contours (only half of it). I-J) Embryos at 30hpf were double *in situ* hybridized with *atoh1a* (green) and *atoh1b* (magenta) and cell proliferation was assessed by anti-PH3 staining (white). Dorsal views with anterior to the left. I'-J') Reconstructed transverse views of (I-J) at the level pointed by the white arrow in (I-J). Note *atoh1a*-cells underwent mitosis, whereas fewer *atoh1b*-cells did. Dotted line corresponded to the neural tube contour. K-K") Tg[atoh1a:GFP] embryo after anti-PH3 (magenta) and DAPI (blue) staining. K'-K") Reconstructed transverse views of the region framed in (K), which is a dorsal view with anterior to the left. This is an example of an apical atoh1a:GFP cell undergoing division (black asterisk) and lateral atoh1a:GFP cell that did not (white asterisk), with (K') or without (K") the red-PH3 staining. Note that atoh1a:GFP cell nuclei expressing PH3 are located in the apical region (black asterisks), whereas atoh1a:GFP cell nuclei negative for PH3 (most probably *atoh1b*-positive, white asterisk) are in the most lateral domain. L-L") Tg[atoh1a:GFP] embryo incubated for 30min with BrdU (blue) and assayed for *atoh1b in situ* hybridization (magenta). Reconstructed transverse views with (L-L') or without (L") *atoh1b*-staining. White asterisks indicate *atoh1b* cells that did not incorporate BrdU. Dotted line corresponded to the neural tube contour. op, otic placode; ov, otic vesicle; r, rhombomere. Scale bars correspond to 50 μm.

zone facing the brain ventricle after lumen expansion (S1C Fig, 24hpf; compare S2A', S2B', S2E' and S2F' with S2C', S2D', S2G' and S2H' Fig). At 24hpf, *ascl1a/b* expression was restricted to rhombomeres, and by 42hpf their expression was clearly confined to the rhombomeric domains that flank the hindbrain boundaries (S2A–S2D Fig) as previously shown in [32,33]. Finally, *neurog1* was expressed in a more ventral position (S1M–S1O and S1M'–S1O' Fig), just below *ascl1a* (S1S Fig), and its expression restricted to the flanking boundary domains by 42hpf (S2E–S2H' Fig; [32]). Thus, by double *in situ* hybridization experiments we could assess the organization of the different proneural progenitor pools along the DV axis as following: *atoh1a*, *ptf1a/ascl1a*, *ascl1b*, *neurog1*, being *atoh1a*-cells the dorsal most progenitor cell population (S1P–S1S Fig). Interestingly, this was not the same order than proneural gene expression in the zebrafish spinal cord, where a second domain of *neurog1* progenitors positioned just underneath the *atoh1a* domain [34]. Proneural genes were expressed in non-differentiated progenitors, and accordingly, non-overlapping expression was observed with HuC-staining (S2A'–S2H' Fig; S3A', S3B and S3C Fig). Interestingly, progenitors located in the dorsal most domain, became placed more lateral upon morphogenesis (see *atoh1a*-expressing cells in Fig 1E and 1E'; S3A' Fig); and progenitors in the ventral region such as *neurog1*-cells, ended up in a more medial position (S2E'–S2H' Fig), showing the impact -and therefore the importance- of morphogenetic changes in the allocation of progenitor cells.

## *atoh1a* and *atoh1b* were sequentially expressed in partially overlapping domains

The three *atoh1* paralogs -*atoh1a*, *atoh1b* and *atoh1c*- were shown to be expressed within the hindbrain and to contribute to the development of the cerebellum, with the expression of *atoh1c* restricted to the upper rhombic lip [17,18]. Since our main interest was understanding the development of the lower rhombic lip (LRL), we focused on the study of *atoh1a* and *atoh1b* and compared their onset of expression. *atoh1a* preceded the expression of *atoh1b* in the most dorsal progenitor cells of the hindbrain at 14hpf (Fig 1A and 1A'). This was in contrast with the onset in the otic epithelium, where *atoh1b* was expressed earlier than *atoh1a* (see magenta in the otic placode in Fig 1A; [27]). At 18hpf, *atoh1a* expression remained in the dorsal most cells, whereas *atoh1b* expression domain was more lateral, overlapping with *atoh1a*-cells and mostly contained within this expression domain (Fig 1B, 1B', 1C and 1C'). Upon the opening of the neural tube, the *atoh1a/b* domains were laterally displaced and *atoh1a* remained medial whereas *atoh1b* positioned lateral (Fig 1D and 1D'), and by 42hpf -when the fourth ventricle was already formed- *atoh1b* expression was completely lateral, and *atoh1a* remained dorsal and medial (Fig 1E and 1E'). Thus, *atoh1a* and *atoh1b* were dorsally expressed but they differed in their mediolateral (apicobasal) position. To demonstrate that they were kept as progenitor cells, we stained Tg[HuC:GFP] embryos with *atoh1a/b* and observed that neither *atoh1a* nor *atoh1b* were expressed in differentiated neurons (Fig 1F–H and 1F'–1H'). Their differential apicobasal distribution and the fact that progenitor cell divisions always happened in the apical domains, suggested that *atoh1b*-progenitor cells might have experienced a basal displacement of their cell body before undergoing differentiation. To demonstrate this, we stained embryos with *atoh1a/b* and anti-pH3, a marker for mitotic figures, and observed that more *atoh1a* than *atoh1b* cells seemed to undergo mitosis (Fig 1I, 1I', 1J and 1J'). In this same line, analyses of single mitotic cells in the transgenic Tg[atoh1a:GFP] fish line that labeled *atoh1a*-expressing cells and their derivatives [18], showed that mitotic atoh1a:GFP cells were always located in the ventricular domain (Fig 1K–1K"; see black asterisks in Fig 1K' and 1K"), whereas the ones that did not divide were laterally displaced just above the neuronal differentiation domain (see white asterisks in Fig 1K' and 1K") as *atoh1b* cells. To demonstrate

that indeed basal *atoh1b* did not proliferate, embryos were incubated with BrdU and assayed for *atoh1b* expression (Fig 1L–1L"). We observed that indeed *atoh1b* cells did not incorporate BrdU, and therefore did not undergo S-phase (see white asterisks in Fig 1L–1L"). Thus, *atoh1b* cells may derive from *atoh1a* progenitors that diminished their proliferative capacity and behaved as committed progenitors transitioning towards differentiation.

### *atoh1a* progenitors gave rise to *atoh1b* cells and *lhx2b* neurons

Next, we sought to unravel whether *atoh1b* cells derived from *atoh1a* progenitors and to which neuronal derivatives the *atoh1a* progenitors gave rise. For this we used the same Tg[atoh1a: GFP] fish line than before [18], which allows to label the cell derivatives of *atoh1a* progenitors due the stability of GFP, and combined *in situ* hybridization experiments with immunostaining, using *atoh1* probes and specific neuronal differentiation genes such as *lhx2b*, *lhx1a*, and pan-neuronal differentiation markers such as HuC (Fig 2; S3 Fig). Although neuronal progenitors expressing *atoh1a* were restricted to the dorsal most region of the hindbrain, their derivatives were allocated in more ventral domains already at early stages of neuronal differentiation (Fig 2A and 2A', compare magenta and green domains). *atoh1b* cells, located more laterally than *atoh1a* cells, expressed GFP (Fig 2B and 2B', see white arrowhead in B' pointing to magenta/white cells in the green territory) indicating that indeed, they derived from *atoh1a* progenitors and according to their position they were transitioning towards differentiation. At this stage in which neuronal differentiation just started, ventral *atoh1a* derivatives constituted a lateral subgroup of differentiated neurons expressing the terminal factor *lhx2b* (see white asterisks indicating magenta/white cells in Fig 2C and 2C'). Note that the more medial *lhx2b* neurons in r4 did not arise from *atoh1a* cells (Fig 2C, see white arrowhead, and compare it with D). This was expected because the lateral domain of *lhx2b* cells always fell below the *atoh1a* progenitors (S3A' Fig), when compared to the more medial domain falling underneath *ascl1b* cells (S3A' and S3B Fig). When the pan-neuronal differentiation marker HuC was analyzed (Fig 2E and 2F), we could clearly observe that at these early stages *atoh1a* derivatives contributed to a portion of differentiated cells (compare Fig 2E and 2E', with 2F and 2F'). Thus, the Tg[atoh1a:GFP] line labeled several cell populations: i) two progenitor cell pools -the one expressing *atoh1a*, and another expressing *atoh1b*-, and ii) the lateral domain of differentiated *lhx2b* neurons. By 48hpf, most of the *atoh1a* progenitors have differentiated, and the remaining *atoh1a/b* progenitor pools were very small (Fig 2G, 2H, 2G' and 2H'). Although *lhx2b* neurons occupied two territories, one lateral and one medial (see white asterisk and arrowhead in S3A and S3A' Fig, respectively), the *atoh1a* derivatives specifically contributed to the most laterally located *lhx2b* neurons (see white asterisk pointing to magenta/white cells in Fig 2I and 2I'; see white asterisks in S3A and S3A' Fig) and did not give rise to the medial *lhx2b* neurons (see white arrowhead in Fig 2I and 2I') or *lhx1a* neurons (S3B Fig). Concomitantly to the growth of the HuC-positive mantle zone, the neuronal differentiation domains dramatically increased (see white and magenta domains in Fig 2K, 2K', 2L and 2L', respectively; see green domains in S2C', S2D', S2G' and S2H' Fig). As expected, cells organized properly along the DV axis according to their differentiation state: progenitor cells in the ventricular domain and cells transitioning towards differentiation more ventrally located (S3C–S3C" Fig). To better understand the dynamics of *atoh1a*-expressing progenitors, we *in vivo* monitored how the atoh1a:GFP cells populated the ventral domain of the hindbrain. We observed that the first-born *atoh1a* neurons occupied the rhombomeric edges or boundary regions (see white arrowhead in S4A–S4C Fig; Fig 2D). By 48hpf, *atoh1a*-derivatives already populated the basal domain of the hindbrain (which it is ventrally located at this morphogenetic stage), generating arched-like structures that coincided with rhombomeric boundaries (see yellow arrowhead in

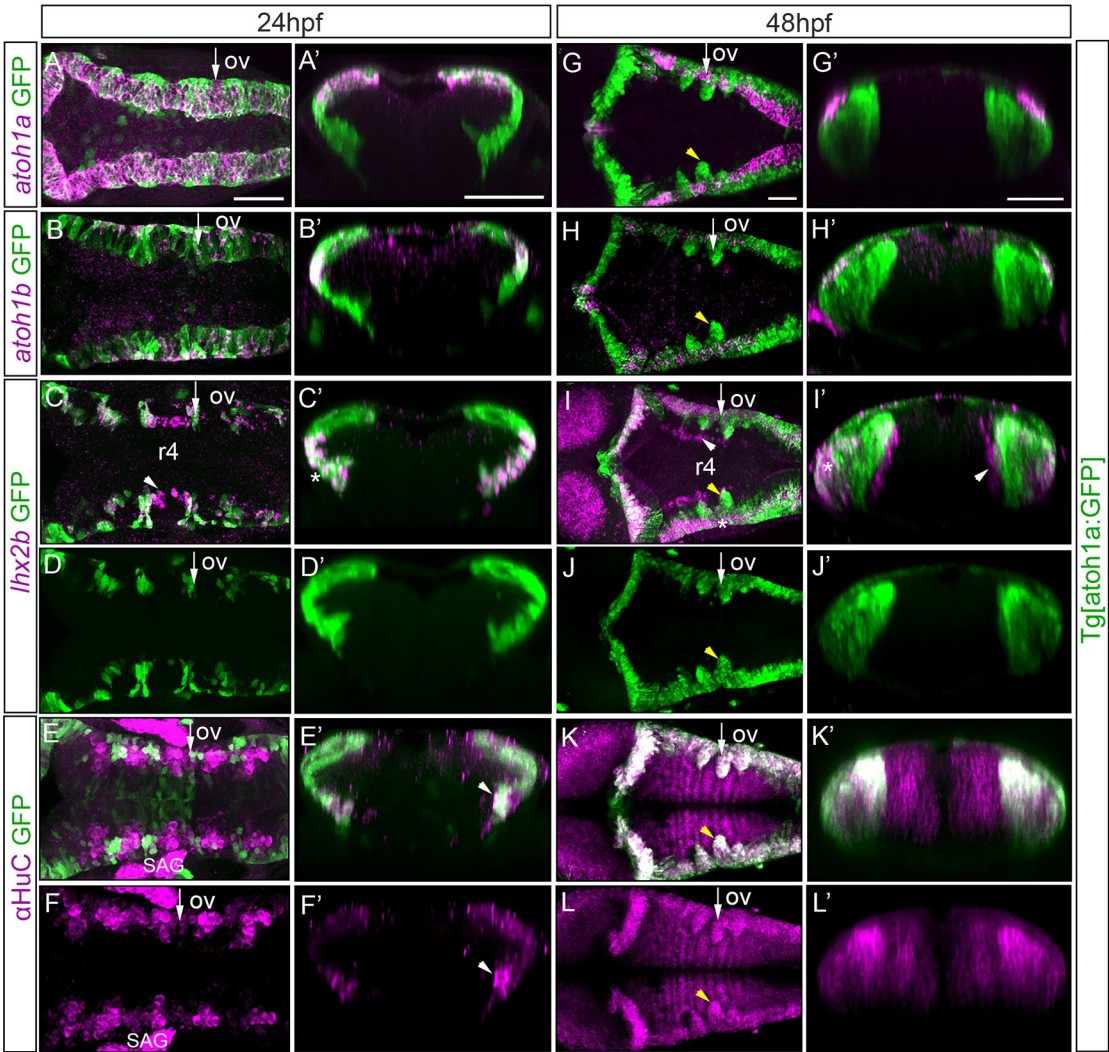

**Fig 2. Analysis of the *atoh1a* neuronal derivatives in Tg[atoh1a:GFP] embryos.** Tg[atoh1a:GFP] embryos at 24hpf and at 48hpf were assayed for *atoh1a* (A, G), *atoh1b* (B, H), *lhx2b* (C-D, I-J) *in situ* hybridization, and anti-HuC (E-F, K-L) staining. Dorsal views of confocal MIP from dorsal stacks (A-B) or ventral stacks (C-L) with anterior to the left. A'-L') Reconstructed transverse sections of the dorsal views in (A-L) at the level indicated with the white arrow depicted in (A-L) corresponding to r4/r5. All embryos displayed the *atoh1a*-progenitors and derivatives in green. Note that *atoh1b* cells derive from atoh1a:GFP progenitors (B', H'), as well as the lateral *lhx2b* neuronal domain (see white asterisks in C', I-I'), whereas the medial *lhx2b* neuronal column in r4 is devoid of green staining (see white arrowhead in C, I-I'). See that differentiated neurons organize in arch-like structures (yellow arrowhead in G-L). ov, otic vesicle; SAG, statoacoustic ganglion; r, rhombomere. Scale bars correspond to 50 μm.

Fig 2G–2L, see white arrowheads in S4 Fig), implying that once the dorsal progenitors commit, they undergo cellular migration during differentiation.

In summary, *atoh1a* progenitors gave rise to *atoh1b* cells and to the lateral domain of *lhx2b* neurons. First differentiated *atoh1a* cells placed between rhombomeres to finally populate the basal hindbrain and generate arched-liked structures.

## Reconstruction of the *atoh1a* lineage

Next question was to address how the rate of differentiation and proliferation of *atoh1a* cells was balanced to achieve the needed cell diversity. For this, we used genetic lineages that allowed to delineate cell types arising from *atoh1a* subsets. To trace the *atoh1a* neuronal

lineages we used a transgenic line that expressed the H2A-mCherry fluorescent reporter protein under the control of enhancer elements of the *atoh1a*. Tg[atoh1a:H2A-mCherry] fish were crossed with Tg[CAAX:GFP] -to have the contour of the cells- and embryos at 24hpf were imaged over 14h. Information about plasma membrane, cell fate and position was simultaneously recorded every 7min (Fig 3A as an example). We monitored the *atoh1a* progenies and studied their behavior according to their position along the DV axis to (Fig 3B–3E). We tracked 40 *atoh1a*-cells, 22 dorsal most (see cells encircled in orange in Fig 3B) and 20 adjacently ventral (see cells encircled in white in Fig 3C), and analyzed their trajectories, when and how many times they divided during the 14h that they were imaged (Fig 3D), and by which mode of division they did so (Fig 3E) attending to their morphology and location: symmetrically giving rise to two progenitor cells (PP) or two neurons (NN), or asymmetrically generating one progenitor cell and one neuron (NP). Of the 22 tracked dorsal most cells (Fig 3B and 3D), only 59% of them divided, and they did so only once (Fig 3D, orange bars; n = 13/22). On the other hand, 82% of cells located just in the underneath domain underwent cell division either once or twice (Fig 3C and 3D, white bars; n = 14/17). Dorsal most *atoh1a* cells undergoing division gave rise always to two cells ending up as differentiated neurons (Fig 3E, dorsal cells NN n = 13/13), whereas the *atoh1a* cells located just below divided according to the three modes of division: 35% gave to two progenitor cells (Fig 3E, ventral cells PP n = 7/20) or two differentiated neurons (Fig 3E, ventral cells NN n = 7/20), and 30% displayed an asymmetric division (Fig 3E, ventral cells NP n = 6/20). These results demonstrated that the dorsal most domain allocated *atoh1a* cells already transitioning towards differentiation, whereas the proliferating *atoh1a*-progenitor pool occupied the region just underneath, generating a dorsoventral gradient of neuronal differentiation.

## *atoh1a* is necessary and sufficient for neuronal specification

Our observations suggested that proliferating *atoh1a* progenitors gave rise to post-mitotic *atoh1b* precursors and *lhx2b* neurons in a sequential manner. However, in order to elucidate the hierarchy between these factors and cellular types, we analyzed the effect of *atoh1a* mutation on the neuronal differentiation domain (Fig 4). We made use of the available *atoh1a*$^{fh282}$ mutant fish in the Tg[atoh1a:GFP] background, which carried a missense mutation within the DNA-binding domain [18]. First, we observed that mutation of *atoh1a* resulted in a complete loss of *atoh1b* expression within the hindbrain (Fig 4A, 4A', 4D, 4D', 4G, 4G', 4J and 4J'), suggesting that *atoh1a* was necessary for *atoh1b* expression and supporting the previous result that *atoh1b* cells derived from *atoh1a* progenitors. This phenotype was accompanied with the loss of the most lateral *lhx2b*-neuronal population (see white asterisk in Fig 4B, 4B', 4E, 4E', 4H, 4H', 4K and 4K'), but not of the *lhx2b*-medial column in r4 that remained unaffected (see white arrowhead in Fig 4B, 4B', 4E, 4E', 4H, 4H', 4K and 4K'), as it was anticipated since this specific population of *lhx2b* neurons did not derive from the *atoh1a* cells (Fig 2D). Although the overall pattern of neuronal atoh1a:GFP cells was not dramatically changed (Fig 4C, 4C', 4F, 4F', 4I, 4I', 4L and 4L'), when the number of neurons at different AP positions was assessed we could observe a clear decrease in the number of differentiated *atoh1a* neurons in the *atoh1a*$^{fh282}$ mutant embryos at both the onset and progression of neuronal differentiation (Fig 4M and 4N, quantification of green dashed inserts in Fig 4C, 4F, 4I and 4L; Table 1).

To address the possibility that the decrease in the number of neurons in *atoh1a*$^{fh282}$ mutants was the result of a smaller number of *atoh1a* progenitor cells, we quantified the number of LRL atoh1a:GFP cells undergoing mitosis (Fig 5A), and the overall number of atoh1a:GFP cells (Fig 5B), both in *atoh1a*$^{WT}$ and *atoh1a*$^{fh282}$ embryos. No significative differences were observed, suggesting that loss of atoh1a function did not affect the original number of LRL

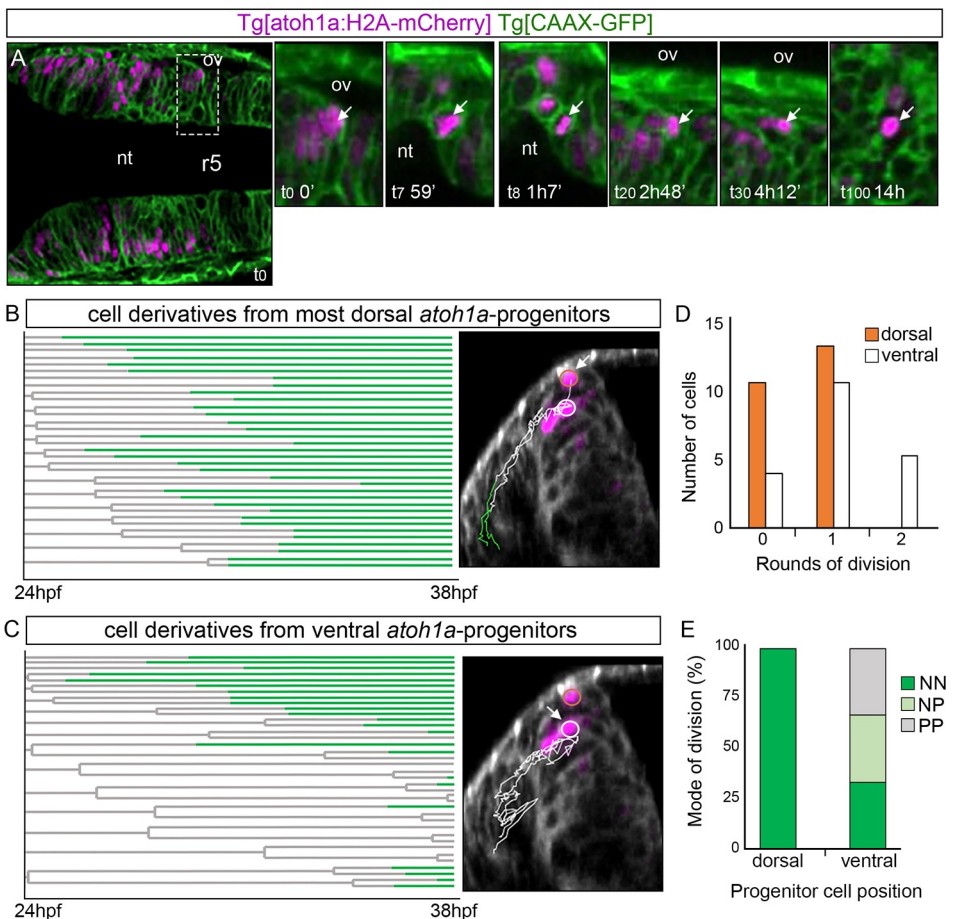

**Fig 3. Cell lineages and behavior of *atoh1a*-derivatives.** Tg[atoh1a:H2A-mCherry] Tg[CAAX:GFP] embryos were imaged from 24hpf during 14h, and information about cell position was acquired every 7min. A) Dorsal view of an embryonic hindbrain displaying *atoh1a* cells in magenta with anterior to the left. The inserts display magnified stills from the framed area in (A) at different times (see white arrow as example of a cell that was tracked from t0 to t100). Note the cell nucleus displacement towards the apical side before division (t8). B-C) Cell lineages from r4 and r5 *atoh1a*-progenitors located at different dorsoventral levels within the *atoh1a* domain; n = 22 in (B) and n = 17 in (C). Each line corresponds to a single cell that branches upon division. Lines are colored according to cell differentiation status: progenitors in grey and differentiated cells in green. The X-axis corresponds to developmental time. The right-hand images display examples of the trajectories of the *atoh1a* tracked cells (white arrow) on the top of the transverse views at t0 (24hpf). Cell trajectories are color-coded according to cell differentiation status: progenitors are in white and differentiated cells in green. Cells are considered differentiated neurons when they are within the neuronal differentiation domain. Dorsal most *atoh1a* cells are encircled in orange and ventral *atoh1a* cells are encircled in white. D) Histogram displaying the number of most dorsal (orange) or ventral (white) atoh1a:GFP cells that undergo different number of divisions over time. Note that *atoh1a*-cells that are more dorsally located undergo less division rounds (orange bars) than the ones in a more ventral position (white bars). E) Mode of cell division according to the DV position of the *atoh1a*-progenitor cells. NN, progenitors giving rise to two neurons; NP, progenitors generating one neuron and one progenitor; PP, progenitor cells that give rise to two progenitors. Note that most dorsal *atoh1a* cells give rise to differentiated cells in all analyzed cases (n = 22 *atoh1a* progenitors), whereas *atoh1a* cells more ventrally located employ the three modes of division (n = 17 *atoh1a* progenitors). nt, lumen of the neural tube; ov, otic vesicle; r, rhombomere.

progenitors (Fig 5A; LRL atoh1a:GFP cells displaying PH3-staining: *atoh1a*$^{WT}$ 17.9 ± 3.6 cells n = 15 vs. *atoh1a*$^{fh282}$ 15.9 ± 3.1 cells, n = 8; Fig 5B; total atoh1a:GFP cells: *atoh1a*$^{WT}$ 69.5 ± 6.4 cells n = 15 vs. *atoh1a*$^{fh282}$ 68.4 ± 7.5 cells, n = 8; see Table 2). Next, we investigated whether *atoh1a* mutation resulted in an increase of apoptotic cells by TUNEL assay (Fig 5C and 5D). The pattern of cell death was the same sparse staining in the wild type and *atoh1a*$^{fh282}$ sibling

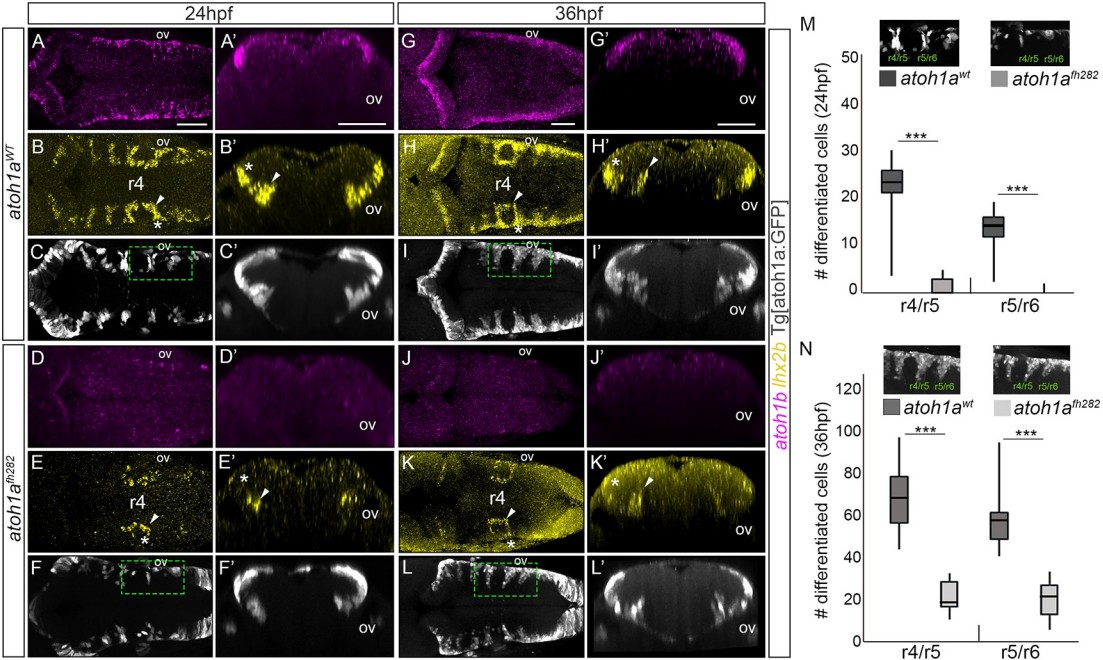

**Fig 4. *atoh1a* is required for the specification of the *lhx2b* neuronal population.** A-L) *atoh1a^{WT}* and *atoh1a^{fh282}* embryos in the Tg[atoh1a:GFP] background were analyzed at 24hpf (*atoh1a^{WT}* n = 14; *atoh1a^{fh282}* n = 18) and 36hpf (*atoh1a^{WT}* n = 11; *atoh1a^{fh282}* n = 7) with *atoh1b* (A, D, G, J), *lhx2b* (B, E, H, K), and anti-GFP in order to follow the *atoh1a*-derivatives (C, F, I, L). A'-L') Reconstructed transverse views of dorsal views displayed in (A-L) at the level of the anterior side of the otic vesicle. Note that *atoh1b* expression (compare A-A' and G-G' with D-D' and J-J'), and the lateral domains of *lhx2b* diminished (compare white asterisks in B-B' with E-E', and H-H' with K-K'), whereas the more medial domain does not decrease so dramatically (compare white arrowheads in B-B' with E-E', and H-H' with K-K'). Note that atoh1a:GFP cells remained, suggesting that there is no massive cell death. M-N) Quantification of differentiated neurons in the r4/r5 and r5/r6 domains of *atoh1a^{WT}* and *atoh1a^{fh282}* embryos as depicted in the small inserts showing dorsal views of halves hindbrains that correspond to the framed regions in (F-L), *** p<0.001 (Table 1 for values and statistical analysis). Note the reduction in the number of atoh1a:GFP differentiated neurons in *atoh1a^{fh282}* embryos. ov, otic vesicle; r, rhombomere. Scale bars correspond to 50 μm.

embryos (Fig 5C and 5D; Table 2), suggesting that mutation of *atoh1a* did not result in a substantial increase of apoptosis. Since the domains of neural bHLH gene expression are established and/or maintained by cross-repression resulting in the control of specific neuronal populations [1], we sought whether this neuronal loss was due to a change in cell fate rather than to a reduction of the number of progenitor cells. Thus, we analyzed proneural gene expression changes both in wild type and mutant context (Fig 5E–5J; *atoh1a^{WT}* n = 8, *atoh1a^{fh282}* n = 10). We observed that upon *atoh1a* mutation, *atoh1a* expression dramatically increased as previously reported [18] (compare Fig 5E and 5E' with 5H and 5H') and the GFP-expressing progenitor cells did not die (Fig 5F–5F", 5I and 5I'). In addition, these cells remained in an intermediate domain since they did not completely migrate towards their final ventral destination as they did in *atoh1a^{WT}* embryos (compare Fig 5F' and 5I'; see white arrow in Fig 5H'–5J'). When we analyzed their possible cell fate switch, by assessing whether the GFP-expressing progenitor cells in the mutant context acquired the expression of the adjacent proneural gene *ptf1a*, atoh1a:GFP progenitors in the *atoh1a^{fh282}* embryos did not display *ptf1a* expression (compare Fig 5G, 5G', 5J and 5J', see white arrow in J'). These observations indicated that in the absence of atoh1a function cells remained as post-mitotic but undifferentiated progenitors, and the LRL domain was properly specified since no changes in the number of cells was observed.

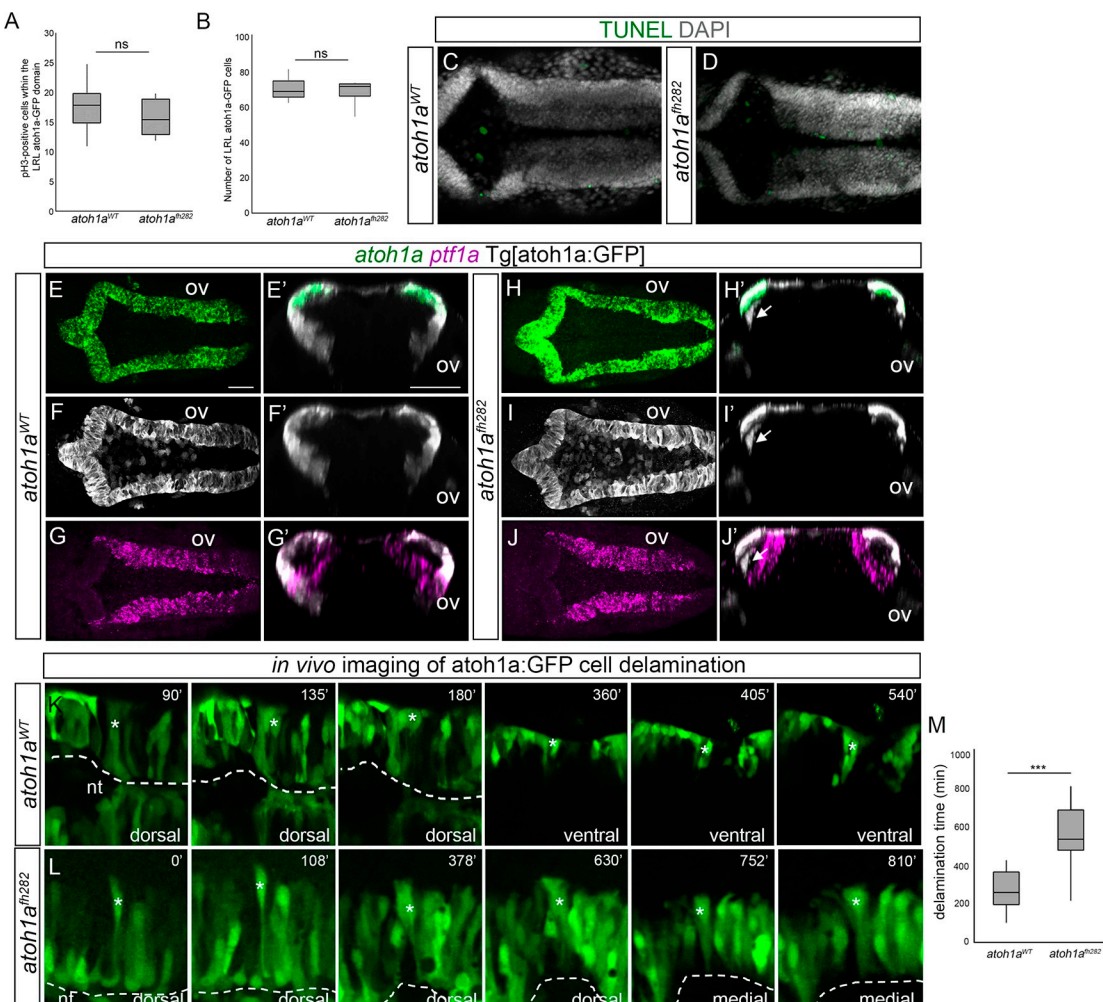

**Fig 5. *atoh1a^{fh282}* mutation does not result in changing the cell fate or cell loss.** A-B) Box-plots with the quantification of mitotic figures within the LRL atoh1a:GFP cells (A), and the total number of LRL atoh1a:GFP cells (B), in *atoh1a^{WT}* and *atoh1a^{fh282}* embryos. C-D) *atoh1a^{WT}* and *atoh1a^{fh282}* embryos in the Tg[atoh1a:GFP] background were analyzed for apoptotic figures by TUNEL. Note that no differences between wild type and mutant embryos were observed (Table 2 for values and statistical analysis). E-G) *atoh1a^{WT}* (n = 8) and (H-J) *atoh1a^{fh282}* (n = 10) embryos in the Tg[atoh1a:GFP] background were concomitantly analyzed for *atoh1a* (E, H), *atoh1a*-derivatives visualized with anti-GFP staining (F, I) and *ptf1a* (G, J) expression. E'-J') Reconstructed transverse views of dorsal views displayed in (E-J) at the level of the otic vesicle. Note that the atoh1a:GFP cells in the *atoh1a^{fh282}* mutant did not migrate towards the differentiation domain and did not display *ptf1a* (see white arrow in H'-J'), indicating that progenitor cells did not switch fate. K-L) Time-lapse stills showing delamination from the LRL of tracked atoh1a:GFP cells (indicated with white asterisk) in *atoh1a^{WT}* (n = 28) and *atoh1a^{fh282}* (n = 12) embryos in the Tg[atoh1a:GFP] background. Dorsal views of hemi-neural tubes (dashed white line indicates the apical region of the hindbrain), with anterior to the left and lateral at the top. Numbers at the top-right indicate the minutes after the beginning of the movie. Note that in wild type embryos, the cell delaminates and migrates towards ventral, allocating in the corresponding neuronal differentiation zone (see the first three dorsal frames and then the following ventral ones), whereas in *atoh1a^{fh282}* embryos the indicated cell remains within the dorsal epithelium (see that there are four dorsal and two medial frames because the cell never reaches ventral). M) Box-plot indicating the time of delamination from the LRL of atoh1a:GFP cells in *atoh1a^{WT}* and *atoh1a^{fh282}* embryos. Note that cells from wild type embryos exit the LRL much earlier than the cells from mutant siblings. Since the *atoh1a^{fh282}* mutant allele only caused a deleterious phenotype in homozygosity, wild type and heterozygous conditions showed identical phenotypes and they were displayed as single wild type condition. nt; neural tube lumen; ov, otic vesicle. Scale bars correspond to 50 μm. ns, non-statistically significant; *** p<0.001.

Loss of atoh1a function resulted in accumulation of atoh1a:GFP progenitors unable to migrate and finally differentiate. In order to demonstrate that these committed precursors arrested, we performed high-resolution time-lapse imaging of both *atoh1a^{WT}* and *atoh1a^{fh282}*

embryos from 24hpf onwards and followed the birth and migration of these atoh1a:GFP progenitors (Fig 5K and 5L). Before migrating, *atoh1a* progenitors in the wild type context, extended their apical and basal feet along the mediolateral axis of the neuroepithelium (dorsal stacks in Fig 5K; white asterisk indicating the tracked cell), and then moved away from the dorsal epithelium towards the mantle zone where they resided as differentiated neurons (see ventral stacks in Fig 5K; white asterisk indicating the tracked cell). This transition was accomplished in an average period of 4.5h (Fig 5K and 5M; t = 275min ± 102; n = 28 tracked cells). In contrast, *atoh1a*$^{fh282}$ progenitors failed to transition and detach (see dorsal stacks in Fig 5L; white asterisk indicating the tracked cell) to barely migrate basally (see medial stacks in Fig 5L; white asterisk indicating the tracked cell). Indeed, after 9.5h of imaging most of *atoh1a*$^{fh282}$ cells still remained in the dorsomedial epithelial region (Fig 5L and 5M; t = 569min ± 180; n = 9/12 tracked cells). Thus, our observations revealed that *atoh1a* was necessary for initial steps of neuronal differentiation (apical abscission and migration).

To further demonstrate the requirement of *atoh1a* in *atoh1b* expression and *lhx2b* neuronal differentiation, and to better dissect the proneural gene hierarchy, we performed conditional gain of function experiments. We injected Mu4127 embryos expressing Gal4 in r3 and r5 with H2B-citrine:UAS vectors carrying either *atoh1a* or *atoh1b* genes, and analyzed the effects in *atoh1* genes and *lhx2b* neurons (Fig 6, Table 3). The *atoh1a* transgene proved successful, as *atoh1a* expression was spread along the DV axis, where it induced the expression of *atoh1b* (compare Fig 6A', 6B', 6D' and 6E') as well as ectopic *lhx2b* neurons in r5 (compare Fig 6C' and 6F'), a rhombomere usually devoid of these neurons at this stage. This was a cell autonomous effect, since all cells expressing *atoh1b* or *lhx2b* ectopically expressed Citrine, and therefore *atoh1a* (compare green cells in Fig 6E–6H with magenta cells in E'-H'). On the other hand, although *atoh1b* expression resulted in ectopic *lhx2b* induction (Fig 6H' and 6I') it did not activate *atoh1a* expression (Fig 5G'), demonstrating that *atoh1b* and *atoh1a* were not interchangeable, and *atoh1a* was upstream *atoh1b*. Overall, our results proved that *atoh1a* progenitors activated *atoh1b*, which allowed them to transition towards differentiation and contribute to the *lhx2b* neuronal population. Moreover, these experiments demonstrated the neurogenic potential of *atoh1b*, and importantly, its role in assigning a neuronal identity subtype.

## Notch-signaling regulates the transition of *atoh1a* cycling progenitors towards *atoh1b* committed cells

We showed that *atoh1a* cycling cells gave rise to *atoh1b* post-mitotic committed precursors. Since this commitment is suspected to be irreversible and leading towards neuronal differentiation, we thought the Notch signaling pathway as a reasonable candidate to be regulating this transition. Thus, we explored the Notch activity within the LRL to understand how *atoh1b* expression was restricted to a given *atoh1a*-domain in the neural tube. First, we assessed Notch activity by the use of the Tg[tp1:d2GFP] transgenic line, which is a readout of Notch-active cells [23]. Indeed, Notch-activity was restricted to the most dorsomedial *atoh1a* cell population (Fig 7A and 7A'), whereas the more laterally located *atoh1b* cells were devoid of it (Fig 7B and 7B'). This suggested that Notch activity was responsible of preventing *atoh1a* progenitor cells to transition to *atoh1b* and therefore modulating neuronal differentiation. To demonstrate this, we conditionally inhibited Notch activity by incubating Tg[atoh1a:GFP] embryos with the gamma-secretase inhibitor LY411575, and asked whether *atoh1a/b* expression domains were altered. Upon inhibition of Notch activity, there was an increase of *atoh1b*-expression at expense of *atoh1a* (Fig 7C, 7D, 7F and 7G): *atoh1b* expression was expanded more medially, and *atoh1a* expression dramatically decreased (compare the border of the *atoh1b* expression in Fig 7D' with 7G'). As expected, the *atoh1b* cells did not arise *de novo* but

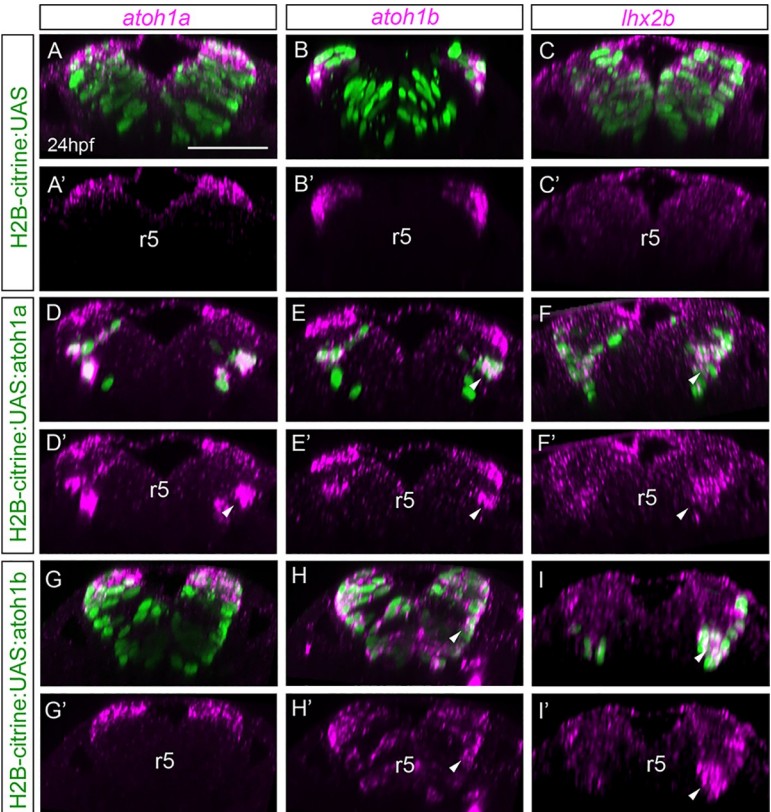

**Fig 6. *atoh1a* is upstream of *atoh1b* and is necessary for *lhxb2* neurons.** Mu4127 embryos expressing Gal4 in rhombomeres 3 and 5 were injected with H2B-citrine:UAS (A-C), H2B-citrine:UAS:atoh1a (D-F) or H2B-citrine:UAS:atoh1b (G-I) constructs in order to ectopically express the gene of interest in r3 and r5. Injected embryos were assayed for Citrine expression (green) and *atoh1a* (A-A', D-D', G-G'), *atoh1b* (B-B', E-E', H-H') or *lhx2b* (C-C', F-F', I-I') expression (magenta). Reconstructed transverse views displaying the merge of the red and green channels (A-I), or only the red channel (A'-I'). Note that ectopic expression of *atoh1a* in more ventral domains induces *atoh1b* and *lhxb2* expression (see white arrowheads in D-F, D'-F'), whereas ectopic *atoh1b* expression induces *lhx2b* but not *atoh1a* (see white arrowheads in H-I, H'-I'). See Table 3 for numbers of analyzed embryos. r, rhombomere. Scale bars correspond to 50 μm.

derived from atoh1a:GFP progenitors (Fig 7E, 7E', 7H and 7H'), supporting the hypothesis that Notch-pathway regulated either the transition from neural stem cells to neuronal progenitors, or the transition of *atoh1a* progenitors towards differentiation. To respond to this question, we conditionally inhibited the Notch-pathway in embryos where *atoh1a* was mutated, and therefore no cells could be transitioning towards differentiation. Upon LY-treatment, *atoh1a^{fh282}* embryos displayed a similar phenotype than non-treated mutant embryos (compare

**Table 3. Analysis of the phenotypes in gain-of-function experiments (Fig 6).**

|  | *atoh1a* | *atoh1b* | *lhx2b* |
|---|---|---|---|
| H2B-citrine:UAS | 16/16 | 13/13 | 18/18 |
| H2B-citrine:UAS:atoh1a | 35/35 | 18/25 | 12/13 |
| H2B-citrine:UAS:atoh1b | 16/16 | 28/28 | 10/14 |

Numbers indicate embryos displaying a phenotype as the one shown in Fig 6, over the total number of analyzed embryos (X/Y).

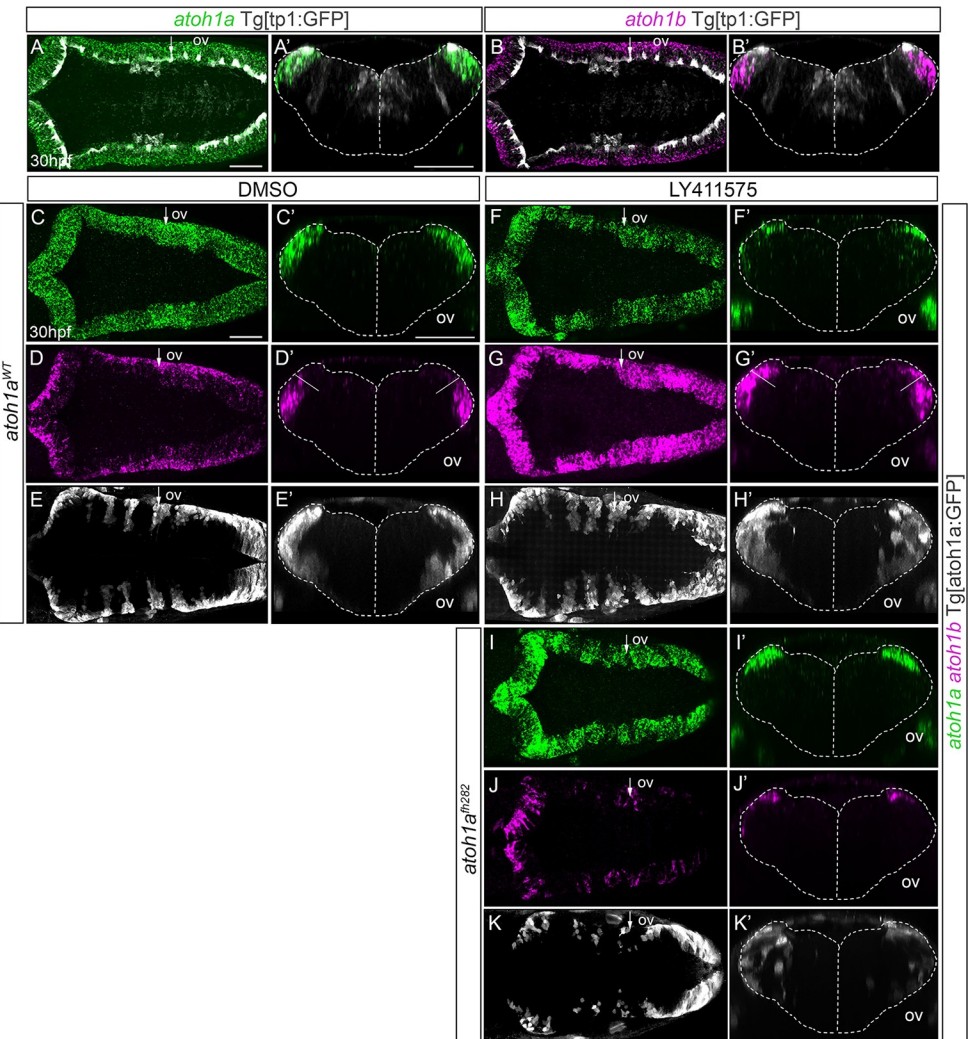

**Fig 7. Notch-signaling regulates the transition of *atoh1a* cycling progenitors towards *atoh1b* committed cells.** A-B) Whole mount double *in situ* hybridization with *atoh1a* (green) and *atoh1b* (magenta) in Tg[tp1:GFP] embryos (readout of Notch-activity in white). A'-B') Reconstructed transverse views of embryos displayed as dorsal views in (A-B) through the point indicated by the white arrow. Note that Notch-activity is restricted to the most dorsomedial tip of the hindbrain, corresponding with *atoh1a* cells. C-K) *atoh1a*[WT]Tg[atoh1:GFP] (C-H) and *atoh1a*[fh282]Tg[atoh1: GFP] (I-K) siblings were double *in situ* hybridized with *atoh1a* (green) and *atoh1b* (magenta) after treatment with DMSO (C-E, n = 10) or the gamma-secretase inhibitor LY411575 (F-H, n = 15; I-K n = 3). The *atoh1a* derivatives were followed by anti-GFP staining in white. C'-K') Reconstructed transverse views of embryos displayed as dorsal views in (C-K) at the level indicated by the white arrow. Note how the *atoh1b*-domain expands at expense of *atoh1a* progenitors after blocking Notch-activity in wild type embryos, but not in *atoh1a*[fh282] mutants. A-D, F-G, I-J) Dorsal views of confocal MIP from dorsal stacks with anterior to the left. E, H, K) Dorsal views of confocal MIP from ventral hindbrain with anterior to the left. ov, otic vesicle. Scale bars correspond to 50 μm.

Fig 7I–7K to Figs 4 and 5), namely: *atoh1a* expression increased (Fig 7I and 7I'; [18]), *atoh1b* expression was highly diminished (Fig 7J and 7J'), and GFP-expressing progenitor cells failed to reach the neuronal differentiation domain (Fig 7K and 7K'). Thus, even though inhibition of N-activity triggered the neurogenic program, lack of atoh1a function impeded the LRL-progenitors to proceed towards differentiation, supporting the hypothesis that the transition of *atoh1a* progenitors towards differentiation depends on atoh1a function and is regulated by Notch.

## Discussion

Progenitor cell populations undergo important changes in their relative spatial distribution upon morphogenesis, which need to be precisely coordinated with the balance between progenitor cells vs. differentiated neurons. Here, we have defined the role of *atoh1* genes along the development of the LRL population, and how this progenitor cell population behaves during the early neurogenic phase.

The spatiotemporal activation of proneural genes in the hindbrain shows that the neurogenic capacity is regionalized along the AP axis, such as that hindbrain boundaries and rhombomere centers remain devoid of neurogenesis [33]. This is valid for most of proneural genes except for *atoh1* genes, because these are expressed all along the AP axis in the dorsal most hindbrain; however, RL derivatives delaminate from the dorsal epithelium, migrate and transitorily locate in the boundary regions. Interestingly, our results demonstrate that the function of different *atoh1* genes depends on the context. In the inner ear, *atoh1a* and *atoh1b* cross-regulate each other but are differentially required during distinct developmental periods: *atoh1b* activates *atoh1a* early, whereas in a late phase *atoh1a* maintains *atoh1b* [27]. In the URL, *atoh1a* and *atoh1c* have equivalent function in the generation of granular cells progenitors [18], whereas we argue that in the LRL *atoh1a* and *atoh1b* are not interchangeable, since they work directionally and have distinct functions. Although in the URL *atoh1a* activates the expression of *neurod1* in intermediate, non-proliferative precursors [35], *neurod1* expression is not detected in the zebrafish LRL before the 48hpf, implying that *atoh1b* is the one defining LRL intermediate precursors rather than *neurod1* during early LRL-derived neurogenesis.

Zebrafish has three *atoh1* genes, *atoh1a*, *atoh1b* and *atoh1c*, which are expressed in overlapping but distinct progenitor domains within the rhombic lip [17,18]. Although *atoh1a* and *atoh1c* specify different, non-overlapping pools of progenitors within the URL, in the LRL while *atoh1b* largely overlaps with *atoh1a* it defines a cellular state rather than a progenitor lineage. *atoh1b* is expressed in a cell population that derives from *atoh1a* progenitors, and it has diminished its proliferative capacity; thus, *atoh1b* cells experienced a basal displacement of their cell body behaving as committed progenitors transitioning towards differentiation. This observation implies that *atoh1* gene duplication in teleosts resulted in a gene sub-functionalization: *atoh1a* may behave as the cell fate selector gene, whereas *atoh1b* functions as a neuronal differentiation gene maintaining the transcriptional program initiated by *atoh1a*. In our conditional functional experiments, *atoh1a* ectopic expression was rapidly downregulated, whereas ectopic *atoh1b* remained active at later stages, highlighting the different roles of *atoh1a* and *atoh1b* in initiating vs. maintaining the differentiation program, and that *atoh1a* and *atoh1b* are not interchangeable. Interestingly, atoh1a/b/c proteins are conserved in the basic region, characterized by being arginine-rich, and in the two helixes but not in the loop, which is known to be variable. This conserved region, the core of bHLH proteins, is located in the center of the three proteins. The N- and C-terminal regions are highly divergent except for certain amino acids such as serine and threonine, predicted to be phosphorylation sites that may modulate the function of the distinct atoh1 proteins (S5 Fig).

Interestingly, first-born neurons from the LRL delaminate and migrate towards medio-ventral positions to allocate in rhombomeric boundaries. Later-born LRL neurons follow the same trajectory, pile up with them and settle more laterally generating what we call neuronal arch-like structures. We think that this pattern of neuronal organization responds to some kind of chemo-attractant signal derived from boundary cells, as first *atoh1a* derivatives have a tendency to allocate within rhombomeric boundaries independently from their AP position upon differentiation. Many of such signalling pathways have been described for LRL migrating cells in the mouse embryo [36]; however, signals participating in this particular context are

unknown. Nonetheless, boundary cells are signalling centres instructing the neuronal alloca-tion in the neighbouring tissue [9]; thus, one plausible hypothesis is that boundary cells might dictate the allocation of newly-differentiated neurons.

Balancing the rate of differentiation and proliferation in developing neural tube is essential for the production of appropriate numbers and achieving the needed cell diversity to form a functional central nervous system (CNS). This requires a finely tuned balance between the dif-ferent modes of division that neural progenitor cells undergo [37]. Three distinct modes of divisions occur during vertebrate CNS development: self-expanding (symmetric proliferative, PP) divisions ensure the expansion of the progenitor pool by generating two daughter cells with identical progenitor potential, self-renewing (asymmetric, PN) divisions generate one daughter cell with the same developmental potential than the parental cell and another with a more restricted potential, and self-consuming (symmetric terminal neurogenic, NN) divisions generate two cells committed to differentiation, thereby depleting the progenitor pool [37,38]. Our *in vivo* cell lineage studies shed light into this specific question in respect to the *atoh1a* cell population. We reveal the importance of the initial allocation of *atoh1a* progenitors: dorsal most *atoh1a* progenitors display more neurogenic capacity than ventral ones, since they give rise only to NN divisions upon the early neurogenic phase, whereas *atoh1a* progenitors located just underneath undergo the three distinct modes of division ensuring the expansion of the *atoh1a*-pool and providing committed progenitors. Most probably, the originally located dor-sal progenitors will quickly become *atoh1b* and transition towards differentiation allocating more laterally. Interestingly, in the amniote spinal cord the modes of progenitor division are coordinated over time [39], instead of space. Why such a difference? One explanation is that in the LRL, where the position of progenitor cells changes dramatically over time, the most effi-cient way to provide fast neuronal production without exhausting the pool of progenitors could be regionalising the proliferative capacity. On the other hand, *in vivo* experiments in the chick spinal cord showed that an endogenous gradient of SMAD1/5 activity dictated the mode of division of spinal interneuron progenitors, in such a way that high levels of SMAD1/5 sig-nalling promoted PP divisions, whereas a reduction in SMAD1/5 activity forced spinal progen-itors to reduce self-expanding divisions in favour of self-consuming divisions [40]. This would suggest that dorsal most *atoh1a* cells would respond less to BMP signalling than ventral *atoh1a* cells. However, during hindbrain morphogenesis there is an important change in the position of *atoh1a* progenitors, and therefore their relative position in respect to the gradient sources. Since morphogen gradients quickly decrease with distance [41,42], it is difficult to apply the same rationale here than in the spinal cord. Still very little is known about how these gradients are established within the hindbrain [43], and how hindbrain progenitors interpret the quanti-tative information encoded by the concentration and duration of exposure to gradients. An alternative explanation is that different E proteins may control the ability of *atoh1a* to instruct dorsal or ventral neural progenitor cells to produce specific, specialized neurons, and thus ensure that the distinct types of neurons are produced in appropriate amounts as it happens in the chick spinal cord [44].

The loss of *atoh1a* function clearly affects the formation of the lateral column of *lhx2b* dif-ferentiated neurons and decreases the number of overall differentiated neurons. But what are the derivatives of these *atoh1a*-derived *lhx2b* cells? It has been described that the hindbrain displays a striking organization into transmitter stripes reflecting a broad patterning of neu-rons by cell type, morphology, age, projections, cellular properties, and activity patterns [45]. According to this pattern, the lateral *lhx2b* column would correspond to glutamatergic neu-rons expressing the *barhl2* transcription factor [46], which in turn is an *atoh1a* target [46,47]. Moreover, our observations revealed that *atoh1a* was necessary for initial steps of neuronal dif-ferentiation, such as apical abscission and migration. Interestingly, this phenotype resembled

to the one of *atoh1c^{fh367}* mutants, in which the release of granule neuron progenitors from the URL required functional atoh1c [18], indicating that *atoh1a* replaced *atoh1c* function in this context.

Notch has been extensively studied as a regulator of proneural gene expression by a process called lateral inhibition, in which cells expressing higher levels of proneural genes are selected as "neuroblasts" for further commitment and differentiation, while concomitantly maintaining their neighbors as proliferating neural precursors available for a later round of neuroblast selection [48]. Indeed, in the LRL the transition *atoh1a* to *atoh1b* seems to be regulated by Notch-activity, since upon Notch-inhibition most of the *atoh1a* cells disappear and they become *atoh1b*, and therefore are ready to undergo differentiation. Thus, although *atoh1a* is the upstream factor in LRL cell specification, several mechanisms seem to be in place to precisely coordinate acquisition of the neurogenic capacity and progenitor vs. differentiation transitions.

## Supporting information

**S1 Fig. Proneural gene expression within the zebrafish embryonic hindbrain.** Whole mount *in situ* hybridization at 18hpf, 21hpf and 24hpf using *atoh1a* (A-C, Q), *ptf1a* (D-F, P), *ascl1a* (G-I, P-S), *ascl1b* (J-L, R) and *neurog1* (M-O, S) probes. Dorsal views with anterior to the left. A'-O') Transverse views at the level pointed by the black arrowhead of embryos displayed in (A-O). P-S) Transverse views of double *in situ* hybridized embryos with the indicated probes. ov, otic vesicle; r, rhombomere.
(TIF)

**S2 Fig. Expression of *ascl1b* and *neurog1* proneural genes along the dorsoventral axis in the context of the neuronal differentiation domain.** Tg[HuC:GFP] embryos were *in situ* hybridized with *ascl1b* (A-D) or *neurog1* (E-H) from 24hpf until 48hpf. A-H) Dorsal views with anterior to the left; A'-H') Reconstructed transverse views at the level pointed by the white arrow in (A-H). Note that progenitor domain in magenta diminishes in size and constitutes the ventricular zone as neuronal differentiation increases over time. ov, otic vesicle. Scale bars correspond to 50 μm.
(TIF)

**S3 Fig. Comparison of the progenitor and differentiated domains upon morphogenesis.** Tg [HuC:GFP] embryos were *in situ* hybridized either with *atoh1a* and *lhx2b* (A-A'), *ascl1b* and *lhx1a* (B), or *ascl1b* and *neuroD4* (C-C"). Reconstructed transverse views except for (A), which is a dorsal view, showing the distinct position of progenitors (*atoh1a* or *ascl1b* in magenta) and differentiated neurons (*lhx2b* and *lhx1a* in green), and cells transitioning towards differentiation (*neuroD4* in green) along the DV axis. ov, otic vesicle; r, rhombomere. Scale bars correspond to 50 μm.
(TIF)

**S4 Fig. First born *atoh1a* cells allocate within the rhombomeric boundaries.** A-E) Double transgenic Tg[atoh1a:GFP]Mu4127 embryos were *in vivo* imaged at different developmental stages. Dorsal views of confocal MIP from ventral hindbrain with anterior to the left. Note that most of the first born atoh1a:GFP cells (green) at 21hpf position at the rhombomeric boundaries as indicated by the magenta staining in r3 and r5 (see white arrowheads indicating the most ventral atoh1a:GFP derivatives). Later, more atoh1a:GFP cells are generated and populate the whole AP axis (see white asterisks in (B-E)) piling up with the first-born atoh1a:GFP cells (see white asterisks). A'-E', A"-E") Reconstructed transverse views of (A-E) at the level of r4/r5 displaying either the two channels (A'-E') or only the green one (A"-E"). See how the atoh1a:

GFP cells corresponding to *atoh1a*-derivatives end up generating a neuronal arch-like structure (see white arrowheads) as development proceeds. ov, otic vesicle; r, rhombomere. Scale bars correspond to 50 μm.
(TIF)

**S5 Fig. Amino acid sequence comparison of zebrafish atoh1 proteins.** Comparison of zebrafish atoh1a, atoh1b and ato1hc proteins by Multiple Sequence Alignment CLUSTALW (MSA, EMBL-EBI). Sequence conservation (>70%) is displayed at the top as grey blocks with different hues. Amino acids highlighted in green correspond to those that match with the consensus sequence, which is displayed at the top in bold. Note how the three atoh1 proteins are conserved in the central regions and their sequence diverge in the N- and C-terminal domains.
(TIF)

## Acknowledgments

We thank R Köster, B Link, and C Moens who kindly provided us with transgenic lines. We would like to thank L Subirana and M Linares for technical assistance, and the members of Pujades lab for critical insights, in special C Engel-Pizcueta who helped to generate the UAS: H2A-mCherry transgenic line.

## Author Contributions

**Conceptualization:** Ivan Belzunce, Cristina Pujades.

**Data curation:** Ivan Belzunce, Carla Belmonte-Mateos.

**Formal analysis:** Ivan Belzunce.

**Investigation:** Ivan Belzunce, Carla Belmonte-Mateos.

**Methodology:** Ivan Belzunce, Carla Belmonte-Mateos.

**Supervision:** Cristina Pujades.

**Validation:** Ivan Belzunce.

**Writing – original draft:** Cristina Pujades.

**Writing – review & editing:** Ivan Belzunce, Cristina Pujades.

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
