## [Decision Letter · Decision Letter 0]

28 Nov 2019

PONE-D-19-29016

The interplay of atoh1 genes in the lower rhombic lip during hindbrain morphogenesis

PLOS ONE

Dear Dr. Pujades,

Thank you for submitting your manuscript to PLOS ONE. After careful consideration, we feel that it has merit but does not fully meet PLOS ONE’s publication criteria as it currently stands. Therefore, we invite you to submit a revised version of the manuscript that addresses the points raised during the review process.

Both reviewers and I are thankful for the high quality of data. One of the reviewers raises only minor points while the second reviewer asks for additional experiments. Sure, it lies in the hands of the authors to reply to the comments. My comments are just some kind of orientation in this review process.

Reviewer 2 asks for BrDU labeling to mark cells is the S-Phase. I might be helpful to perform a BrDU pulse labeling at one time point (30 hrs hpf) to show the postmitotic character of the atoh1b precursors.

Furthermore, Reviewer 2 raises the interesting aspect of whether loss of atoh1a suppresses cell differentiation or cell death. The reviewer asks specifically for apoptosis. Here, I recommend to compare wildtype and atoh1a mutants at one time point and to perform antibody labeling with a marker for apoptosis. This experiment may show whether substantial apoptosis occurs or not.

Both reviewers show a high interest in the aspect of Notch-activity. A single time point experiment regarding Notch activity in the atoh1a mutant may help to answer the concerns of both reviewers.

We would appreciate receiving your revised manuscript by Jan 12 2020 11:59PM. To enhance the reproducibility of your results, we recommend that if applicable you deposit your laboratory protocols in protocols.io, where a protocol can be assigned its own identifier (DOI) such that it can be cited independently in the future. For instructions see: http://journals.plos.org/plosone/s/submission-guidelines#loc-laboratory-protocols

We look forward to receiving your revised manuscript.

Kind regards,

Robert Blum

Academic Editor

PLOS ONE

Journal Requirements:

'Zebrafish (Dario rerio) were treated according to the Spanish/European regulations for the handling of animals in research. All protocols were approved by the Institutional Animal Care and Use Ethic Committees and implemented according to European regulations (protocol CPC-19-0025). Experiments were carried out in accordance with the principles of the 3Rs. Embryos were obtained by mating of adult fish using standard methods.'

Please amend your current ethics statement to include the full name of the ethics committee that approved your specific study.

For additional information about PLOS ONE submissions requirements for animal ethics, please refer to http://journals.plos.org/plosone/s/submission-guidelines#loc-animal-research.

Reviewers' comments:

Reviewer's Responses to Questions

**Comments to the Author**

1. Is the manuscript technically sound, and do the data support the conclusions?

Reviewer #1: Yes

Reviewer #2: Yes

2. Has the statistical analysis been performed appropriately and rigorously? 

Reviewer #1: Yes

Reviewer #2: Yes

3. Have the authors made all data underlying the findings in their manuscript fully available?

Reviewer #1: Yes

Reviewer #2: Yes

4. Is the manuscript presented in an intelligible fashion and written in standard English?

Reviewer #1: Yes

Reviewer #2: Yes

5. Review Comments to the Author

Reviewer #1: This manuscript describes the expression of proneural genes in the zebrafish hindbrain with specific focus on atoh1 genes atoh1a and atoh1b in the lower rhombic lip. Using a long-lived atoh1a reporter transgenic line and an atoh1a mutant as well as atoh1a and atoh1b gain-of-function the authors show that atoh1a is required for the generation of atoh1b-expressing committed progenitors and subsequently to lhx2b+ neurons in the ventro-lateral hindbrain. By live imaging they show that atoh1a in the lower rhombic lip is required for the timely release of dorsal progenitors from the ventricular surface to initiate their lateral and ventral migration into post-mitotic domains, similar to what has been recently described in the upper rhombic lip for atoh1c. Finally, using pharmacological inhibition they show that the transition from atoh1a+ to atoh1b+ identity is dependent on Notch signaling.

The data is compelling, and the conclusions are well-supported. I recommend acceptance if the following comments can be addressed.

Specific criticisms:

In Fig. 5 the authors show an interesting phenotype in atoh1a mutants in which progenitors fail to delaminate from the ventricular surface, which they interpret as meaning that in the absence of atoh1a function progenitors are suspended in a post-mitotic but undifferentiated state. This seems like one of the most interesting findings in the paper, as it may give insight into the mechanism by which atoh1 genes control neurogenesis. However the significance of this finding, its relevance to the proposed functions of atoh1 genes in other contexts or species is not discussed further here or in the discussion. Please add a discussion of this interesting finding.

The authors show that Notch inhibition leads to premature differentiation of atoh1a-expressing progenitors (Fig. 7). It would be interesting to know whether this effect depends on atoh1a. A prediction may be that in atoh1a mutants neuronal progenitors will fail to delaminate even in the presence of the Notch inhibitor.

The discussion section is extensive and generally fine, however parts of it raise questions that are in no way addressed by the research. For instance, line 650 initiates a discussion of interkinetic nuclear migration and raises the question of how actin generates forces for nuclear movement and how that relates to atoh1b. Since no role for atoh1b in this migration has been demonstrated, it seems premature to speculate on it.

Minor criticisms:

• The green dashed box in Fig. 4I is in the wrong place

• line 639: should be URL not UPL

• line 651: should it say “…and move toward the basal surface” (not apical)?

Reviewer #2: The manuscript by Pujades and her colleague describes roles of atoh1a and atoh1b in neuronal differentiation from neural progenitors in the lower rhombic lip (LRL) in zebrafish. By in vivo cell lineage tracing and gain/loss-of function of Atoh1a, the authors try to demonstrate that atoh1a is required and sufficient for specification of LRL cells by activating atoh1b; atoh1b functions as a neural differentiation gene, contributing to the lhx2b neuronal population in a Notch dependent manner. Overall quality of the data is high. Most of the conclusion is supported by their experimental evidence. I have several major concerns described below.

Major points:

1. In page 10, line 268, “we stained embryos with atoh1a/b and anti-pH3…..observed that more atoh1a than atoh1b cells seemed to undergo mitosis (Fig 1I-I’, J-J’)….. atoh1b cells may derive from atoh1a progenitors that diminished their proliferative capacity”. In page 14, line 424, “our observations suggested that proliferating atoh1a progenitors gave rise to postmitotic atoh1b precursors”. The pH3 only marks G2/M cells. The BrdU labeling that marks S cells should be done.

2. In page 14, line 430, “mutation of atoh1a resulted in a complete loss of atoh1b expression within the hindbrain”. In page 15 and Fig 6, misexpression of atoh1a resulted in ectopic expression of atoh1b. The data suggest atoh1a is required and sufficient for expression of atoh1b in the LRL. However, it does not necessary mean that atoh1b is required for atoh1a-depedent neuronal differentiation of the LRL cells. If they would like to claim this issue, they should analyze either atoh1b mutant and/or morphants (lhx2b neurons in the mutants and/or morphants; whether atoh1a can induce lhx2b neurons in the absence of atoh1b).

3. In Fig 4, atoh1b-expressing cells and lhx2b-expressing cells were substantially reduced in the atoh1a mutants. Did cell death (apoptosis) occurs? Whether did loss of atoh1a function suppress neuronal differentiation or induce cell death?

3. In page 16, line 477, “upon atoh1a mutation, atoh1a expression dramatically increased as previously reported (compare Fig 5C and F). I wonder why the atoh1a:GFP expression in the ventricular zone did not increase in the atoh1a mutants (Fig. 5D-D’, G-G’),

4. In page 19, line 581, “Notch-activity was restricted to the most dorsomedial atoh1a cell population (Fig 7A-A’), whereas the more laterally located atoh1b cells were devoid it (Fig 7B-B’). Loss of Notch activity reduced atoh1a expression and increased atoh1b expression”. The authors concluded Notch pathway regulated the transition of atoh1a progenitors toward differentiation. However, as the authors mentioned, the Notch signal in only active in the most dorsomedial atoh1a-expressing cells. Considering the known role of Notch signaling in neural differentiation, these Notch activity-high cells could be in a transition from neural stem cells to neuronal progenitors. Notch inhibition might lead to acceleration of differentiation from stem cells to atoh1a and atoh1b cells. Only the observation at a single point in time might only be looking at a process of the accelerated neuronal differentiation. Time course analysis should help.

Minor points:

1. The sentence “We revealed that atoh1a behaves as the cell fate selector gene” is too strong. The manuscript does not provide evidence showing the role of atoh1a in the cell fate selection.

2. In page 12, line 341, “atoh1a-derived neurons” could be misleading. These cells derived from the atoh1a-expressing progenitors.

3. In page 13, section “Reconstruction of the atoh1a lineage” and Fig. 3, the authors categorized cell divisions to PP, NN and NP. How did they determine neurons (Ns)?

4. In page 15, line 439, “a clear decrease in the number of differentiated atoh1a neurons in the atoh1a mutant embryos at both the onset and progression of neuronal differentiation (Fig 4…). I thought that the authors examined atoh1a:GFP neurons (the sentence is misleading). Did the authors examine whether they are differentiated neurons (HuC+) in the mutants?

4. Misexpression of atoh1a and atoh1b strongly indicate that atoh1a and atoh1b are not interexchangeable. The authors should better discuss difference in the structure of Atoh1a and Atoh1b proteins.

6. PLOS authors have the option to publish the peer review history of their article (what does this mean?). If published, this will include your full peer review and any attached files.

Reviewer #1: No

Reviewer #2: No

---

## [Author Response · Author response to Decision Letter 0]

19 Dec 2019

We thank the Reviewers and the Academic Editor for commenting on our paper in order to improve it. We have followed their advices and implemented their suggestions. We improved and modified previously existing figures including new experiments (Figure 1, Figure 5, Figure 7) and incorporated one new figure (Figure S5) to better support our findings. We provide an amended manuscript that I wish it will be suitable for publication in PLOSone.

The responses to the points raised by the Referees are as follows:

Reviewer 1

1. We have discussed further our observation that in the absence of atoh1a function progenitors are suspended in a post-mitotic but undifferentiated state and included in the Discussion section (p26).

2. We have investigated whether the effect of premature differentiation of atoh1a-expressing progenitors upon Notch inhibition depended on atoh1a and included these new results in Fig 7I-K. Briefly, we inhibited Notch-signaling in atoh1afh282Tg[atoh1a:GFP] mutant sibling embryos, and as predicted neuronal progenitors failed to reach the most ventral territories. We discussed this further after describing the results (p21).

3. We followed the suggestion of the Reviewer of deleting the section in the Discussion about interkinetic nuclear migration and how actin may generate forces for nuclear movement and how that relates to atoh1b. 

All minor points have been addressed.

Reviewer 2

1. The Reviewer suggested to include BrdU-incorporation experiments to demonstrate that indeed atoh1b cells were mainly postmitotic.

We followed the Reviewers advice and performed BrdU-incorporation experiments in Tg[atoh1a:GFP] embryos to demonstrate that atoh1b cells did not enter S-phase. We included these experiments in Figure 1L-L’’. 

2. The Referee wants us to address whether atoh1b is required for atoh1a-dependent neuronal differentiation of the LRL cells.

We have followed the suggestion of the Referee and performed LOF experiments for atoh1a or atoh1b. For this, we used translation blocking atoh1a and atoh1b morpholinos (Millimaki et al, Development 2007). As provided in the following figure, LOF of atoh1b in the HuC:GFP background results in a similar phenotype to MO-atoh1a injected embryos. In our opinion. These results are in the same line with the ones provided in the manuscript, strongly suggesting that indeed atoh1b is downstream of atoh1a in the transition of atoh1a-progenitors toward differentiation. 

Fig. Loss-of-function of atoh1a and atoh1b results in a decrease of differentiated neurons. Tg[HuC:GFP] embryos were injected with: (A) control-MO (n = 24), and (B) translation-blocking morpholino for atoh1a (n = 21), or (C) for atoh1b (n = 20), and analyzed at 42hpf for LRL-derived neuronal arch-like structures (see white arrow in A). Differentiated neurons in green. D) Graphical representation of the phenotype expressivity upon morpholino-injection. Note that none of embryos injected with the control MO displayed an aberrant phenotype (n = 24/24). For atoh1a-morphants, 85% of the embryos had no LRL-derived neuronal arch-like structures as the embryo displayed in (B) (n = 18/21), and 15% showed partial defects (n = 3/21). atoh1b-MO injected embryos showed a disperse distribution with 40% unaffected (n = 8/20), 40% with a mild phenotype (n = 8/20) and 20% with a strong phenotype similar to atoh1a-morphants as embryo displayed in (C) (n = 4/20). Scale bars correspond to 50�m.

3. The Reviewer raises the interesting aspect of whether loss of atoh1a suppresses cell differentiation or cell death. 

In our opinion, the comparison of the total number of LRL atoh1a:GFP cells, and the number of them undergoing mitosis in wild type and atoh1a mutant siblings (Fig 5A-B) clearly suggested that apoptosis could not be playing a major role in the decrease of the number of neurons. However, we followed the advice of the Referee and now we provide a comparison of the apoptotic events occurring in wild type and atoh1afh282 mutant embryos. Results show that there is no substantial contribution of apoptosis to the observed mutant phenotype. We described these results in p17 and displayed them in Fig 5C-D and Table 2.

3. The Referee wonders why the atoh1a:GFP expression in the ventricular zone did not increase in the atoh1a mutants. 

As it was previously described in Kidwell et al, Dev Biol 2018, the atoh1afh282

mutation results in the upregulation of the atoh1a mRNA due to the self-regulation of proneural genes transcription. The atoh1a:GFP does not change because it is a genetic tracer of the cells that should be expressing atoh1a but they do not, not of the atoh1a mRNA. 

4. The Reviewer considers that Notch-pathway could regulate either the transition from neural stem cells to neuronal progenitors, or the transition of atoh1a progenitors towards differentiation.

The Reviewer is absolutely right and to respond to this question, we conditionally inhibited the Notch-pathway in embryos where atoh1a was mutated, and therefore no cells in the LRL could be transitioning towards differentiation. Upon LY-treatment, atoh1afh282Tg[atoh1:GFP] embryos displayed a similar phenotype than non-treated mutant embryos, namely: atoh1a expression increased, atoh1b expression was highly diminished, and GFP-expressing progenitor cells failed to reach the neuronal differentiation domain. These experiments suggested that the transition of atoh1a progenitors towards differentiation depends on atoh1a function and is regulated by Notch. These new results are described in p21 and included in Fig 7I-K.

Minor comments 

1. We rephrased the sentence “We revealed that atoh1a behaves as the cell fate selector gene”, which in Referee’s opinion was too strong (p2, p24) and wrote “We revealed that atoh1a may behave as the cell fate selector gene”.

2. We substituted “atoh1a-derived neurons” by “atoh1a-expressing progenitors” in p13.

3. We categorized cell divisions to PP, NN and NP by position, either in the ventricular zone (P), or in the mantle/differentiation zone (N), and explained it properly in the Fig 3 legend.

4. We examined that atoh1a:GFP cells give rise to HuC-neurons in Fig 2. In the atoh1a mutants we did analyze atoh1a:GFP cells by their position in the mantle zone. 

5. We have better discussed the difference in the structure of atoh1a and atoh1b proteins in p24 and included a new figure with the amino acid sequence comparison of all atoh1 proteins in zebrafish to support this part of the discussion (Fig S5).

---

## [Editor Report · Decision Letter 1]

10 Jan 2020

The interplay of atoh1 genes in the lower rhombic lip during hindbrain morphogenesis

PONE-D-19-29016R1

Dear Dr. Pujades,

We are pleased to inform you that your manuscript has been judged scientifically suitable for publication and will be formally accepted for publication once it complies with all outstanding technical requirements.

With kind regards,

Robert Blum

Academic Editor

PLOS ONE

Additional Editor Comments (optional):

Congratulation for this study. Please note, in Table 2 title, apoptotic figure...you may mean 'hallmarks of apoptosis.

A happy year 2020

Robert Blum
---

## [Editor Report · Acceptance letter]

27 Jan 2020

PONE-D-19-29016R1 

The interplay of *atoh1* genes in the lower rhombic lip during hindbrain morphogenesis 

Dear Dr. Pujades:

I am pleased to inform you that your manuscript has been deemed suitable for publication in PLOS ONE. Congratulations! Your manuscript is now with our production department. 

With kind regards,

on behalf of

PD Dr. Robert Blum 

Academic Editor

PLOS ONE